# Age-effects of sport education model on basic psychological needs and intrinsic motivation of adolescent students: A systematic review and meta-analysis

**Jing Dai[1]☉, Jiayong Chen[1]☉, Zijing Huang[1], Yuhuan Chen[1], Yezi Li[2], Jian Sun[3], Duanying Li[3], Min Lu[3]\*, Jiancai Chen📍[4]\***

**1** Digitalized Performance Training Laboratory, Guangzhou Sport University, Guangzhou, Guangdong, China, **2** Suzhou Gaobo Software Technology Vocational College, Suzhou, Jiangsu, China, **3** Sports Training Institute, Guangzhou Sport University, Guangzhou, Guangdong, China, **4** Physical Education Institute, Guangzhou Sport University, Guangzhou, Guangdong, China

☉ These authors contributed equally to this work.
\* gtchenjiancai@163.com (JC); Lm3899@sina.com (ML)

**Data Availability Statement:** All relevant data are within the manuscript and its Supporting information files.

## Abstract

### Objective

This study explores the age effects of the sport education model(SEM) on the impact of basic psychological needs (autonomy, competence, relatedness) and intrinsic motivation (interest, enjoyment, satisfaction) among adolescent students.

### Method

Retrieval of relevant literature from PubMed, Web of Science, Scopus, and China National Knowledge Infrastructure (CNKI). The search period ranged from the starting year to January 7, 2024. Subsequently, literature screening, data extraction, and quality assessment will be conducted, and data analysis will be performed using "Review Manager 5.4" software.

### Result

Overall, SEM has a positive and statistically significant impact on the basic psychological needs (MD = 0.36, 95% CI [0.22, 0.50]) and intrinsic motivation (MD = 0.75, 95% CI [0.58, 0.93]) of adolescent students (P<0.01). Subgroup analysis revealed age effects on the impact of SEM on the basic psychological needs of adolescent students: pre-peak height velocity (PRE-PHV) (MD = 0.39, 95% CI [0.23, 0.56], $I^2$ = 45%, P<0.01), mid-peak height velocity (MID-PHV) (MD = 0.22, 95% CI [0.01, 0.42], $I^2$ = 82%, P<0.05), post-peak height velocity (POST-PHV) (MD = 1.27, 95% CI [0.79, 1.74], $I^2$ = 0%, P<0.01). Similarly, age effects were found for intrinsic motivation: MID-PHV (MD = 0.86, 95% CI [0.62, 1.11], $I^2$ = 68%, P<0.01), POST-PHV (MD = 0.56, 95% CI [0.40, 0.72], $I^2$ = 0%, P<0.01).

**Funding:** Research Project Funded by the Guangdong Provincial Education Science Planning (2022GXJK242): Theoretical and Empirical Study on the Integration of Competitive Sports Education Model into Ideological and Political Education in Higher Education Physical Education Courses. The research project on higher education for the 2022 Annual Plan of the "14th Five-Year Plan" by the Guangdong Higher Education Society(22GZD010): Research on the ideological and political reform of sports education model in higher education institutions based on the competitive sports education model. The funders had no role in study design, data collection and analysis, decision to publish, or preparation of the manuscript.

**Competing interests:** The authors have declared that no competing interests exist.

## Conclusion

The SEM is an effective approach to enhancing the basic psychological needs and intrinsic motivation of adolescent students. However, it exhibits age effects among students at different developmental stages. Specifically, in terms of enhancing basic psychological needs, the model has the greatest impact on POST-PHV students, followed by PRE-PHV students, while the improvement effect is relatively lower for MID-PHV students. The enhancement effect on intrinsic motivation diminishes with increasing age.

## Introduction

The sport education model (SEM) refers to an instructional approach in physical education that combines sound teaching practices and suitable physical activities to offer each student a teaching model that includes role participation and opportunities for sports learning [1]. This model replaces traditional learning units with sports seasons and is based on the theory of game-based education. It uses team cooperation, role-playing, and autonomous learning as vehicles to provide students with authentic and meaningful physical experiences. The aim is to cultivate students into "cultured, enthusiastic, and capable athletes" [1,2].

Research has shown that basic psychological needs and intrinsic motivation are closely related to students' sports participation, self-awareness, learning interest and enthusiasm, and social adaptability in physical education [3–5]. The self-determination theory suggests that meeting three basic psychological needs—autonomy, competence, and relatedness—is crucial for human development [6]. Intrinsic motivation, on the other hand, is the highest level of motivation and an important driving force behind human behavior in various domains [7].

The SEM has been found to have a positive impact on basic psychological needs and intrinsic motivation [6,8,9]. However, previous studies have overlooked the variable of how basic psychological needs and intrinsic motivation may change with age. It has been found that SEM has different effects on the three different basic psychological needs, which may be due to the differences in age groups [10–12]. Intrinsic motivation tends to decrease with age, and children and adolescents are more likely to exhibit intrinsic motivation [13].

Based on the effectiveness of SEM in improving basic psychological needs and intrinsic motivation, as well as the differences in these aspects among adolescent students at different stages of adolescence. Exploring the age effects of SEM on the impact of basic psychological needs and intrinsic motivation in adolescent students is particularly important. Therefore, this study comprehensively collects published experimental literature, with basic psychological needs (autonomy, competence, relatedness) and intrinsic motivation (interest, enjoyment, satisfaction) as outcome measures, to explore the age effects of SEM on the impact of basic psychological needs and intrinsic motivation among adolescent students.

## Survey methodology

### Search strategy

The search databases included PubMed, Web of Science, Scopus and China National Knowledge Infrastructure (CNKI). The search period ranged from the starting year of inclusion in each database to January 7, 2024. English search keywords included sport education model, sports education mode, sport education curriculum and instruction model, sport education, basic needs, intrinsic motivation, autonomy, competence, relatedness, relationship, motivation, enjoy, satisfaction, interest, adolescent, teenager, youth, youngsters, etc. To ensure the accuracy of the search, two researchers independently cross-checked the search terms. In the

#1 "sport education model"[All Fields] OR "sports education mode"[All Fields] OR "sports education model"[All Fields] OR "Sport Education curriculum and instruction model"[All Fields] OR "SEM"[All Fields] OR "sport education"

#2 "adolescent"[All Fields] OR "teenager"[All Fields] OR "youth"[All Fields] OR "youngsters"[All Fields] OR "adolescent students"[All Fields] OR "adolescents"[All Fields] OR "child"[All Fields] OR "children"[All Fields] OR "student"[All Fields] OR "students"[All Fields] OR "female adolescent"[All Fields] OR "male adolescent"[All Fields]

#3 #1 AND #2

#4 "basic needs"[All Fields] OR "basic psychological needs"[All Fields] OR "intrinsic motivation"[All Fields] OR "autonomy"[All Fields] OR "autonomous"[All Fields] OR "competence"[All Fields] OR "competency"[All Fields] OR "relatedness"[All Fields] OR "Relationship"[All Fields] OR "relation"[All Fields] OR "motivation"[All Fields] OR "enjoy"[All Fields] OR "enjoy"[All Fields] OR "interesting"[All Fields] OR "interest"[All Fields]

#5 #3 AND #4

**Fig 1. PubMed literature search strategy.**

case of any discrepancies, a final decision was made by a third researcher. The search strategy utilized a combination of subject terms and free-text terms, and supplementary searches were conducted using citation tracing when necessary. PubMed was used as an example (Fig 1).

## Study selection

The literature inclusion criteria for this meta-analysis were based on the Populations, Interventions, Controls, Outcomes and Study Design (PICOS) format of evidence-based medicine.

The inclusion criteria were as follows: (1)Participants were adolescents aged 10–19 years [14], with pre-peak height velocity (PRE-PHV) defined as 10–12.99 years, mid-peak height velocity (MID-PHV) defined as 13–15.99 years, and post-peak height velocity (POST-PHV) defined as 16–18.99 years based on maturity status[15]. (2) Since the teaching experiments were mostly quasi-experimental studies, this article included all controlled experiments. (3) The intervention method in the experimental group is SEM, while the intervention method in the control group is traditional teaching mode(Following the school's unified arrangement, the classroom adopts the skill-training-game approach, with the teacher leading the course

and designing, developing, and implementing warm-up exercises and skill practice [6,16]). (4) Outcome measures included basic psychological needs (autonomy, competence, relatedness) and intrinsic motivation (interest, satisfaction, enjoyment). (5) The intervention time and frequency were the same for the experimental and control groups.

The exclusion criteria were as follows: (1) Reviews and conference papers. (2) Inability to access the full text or research data; (3) Physiotherapy Evidence Database (PEDro) score less than 4 points.

Two researchers conducted literature screening respectively, and if there were any disagreements, the final decision was made by a third researcher. The final selection process resulted in a total of 16 articles that met the inclusion criteria for this study. Please refer to Fig 2 for a detailed overview of the process.

## Data extraction

All retrieved literature was imported into Endnote X9 software, and duplicates were removed. Two researchers independently read the titles and abstracts of the articles for preliminary screening. In case of disagreement, a third researcher was invited to make the final decision. After finalizing the literature to be included in the meta-analysis, the data were extracted to a Microsoft Excel spreadsheet. All pre-and post-test data were presented as mean ± standard deviation and eventually converted to change values ± standard deviation.

The extraction included:(1) author name and publication year. (2) participant characteristics: gender, age, sample size. (3) Intervention details: intervention method, duration, frequency, outcome measures, and their respective mean and standard deviation.

**Assessment of risk of bias.** Quality assessment was conducted using the PEDro scale[17], which consisted of 11 items: eligibility criteria specified, random allocation, concealed allocation, baseline comparability, participant blinding, therapist blinding, outcome measures, intention-to-treat analysis, between-group statistical comparisons, point estimates, and measures of variability. Each item was scored as 1 if clearly described or 0 if not clearly described. The total score was out of ten points, where the first item, which only affected the external validity of the study, was not included in the total score. A PEDro score of less than 4 indicated poor quality, 4–5 indicated fair quality, 6–8 indicated good quality, and 9–10 indicated high-quality literature [17]. The quality assessment was independently conducted by two researchers, and in case of disagreement, a third researcher made the final decision.

## Statistical analysis

The data analysis was performed using the "Review Manager 5.4" software. The primary effect measure was the change values ± standard deviation before and after intervention. All outcome measures were analyzed using the weighted mean difference (WMD) with a 95% confidence interval as the summary effect measure. The included data in this study were continuous variables, and the mean difference (MD) was used as the unit of measurement. The degree of heterogeneity in the studies was assessed using $I^2$, where $I^2 < 25\%$ indicated negligible heterogeneity, $25\% < I^2 < 75\%$ indicated moderate heterogeneity, and $I^2 > 75\%$ indicated high heterogeneity [17]. If the heterogeneity was less than 25%, a fixed-effects model was used for the analysis. If it exceeded 25%, a random-effects model was used. A significance level of $P < 0.05$ was considered statistically significant.

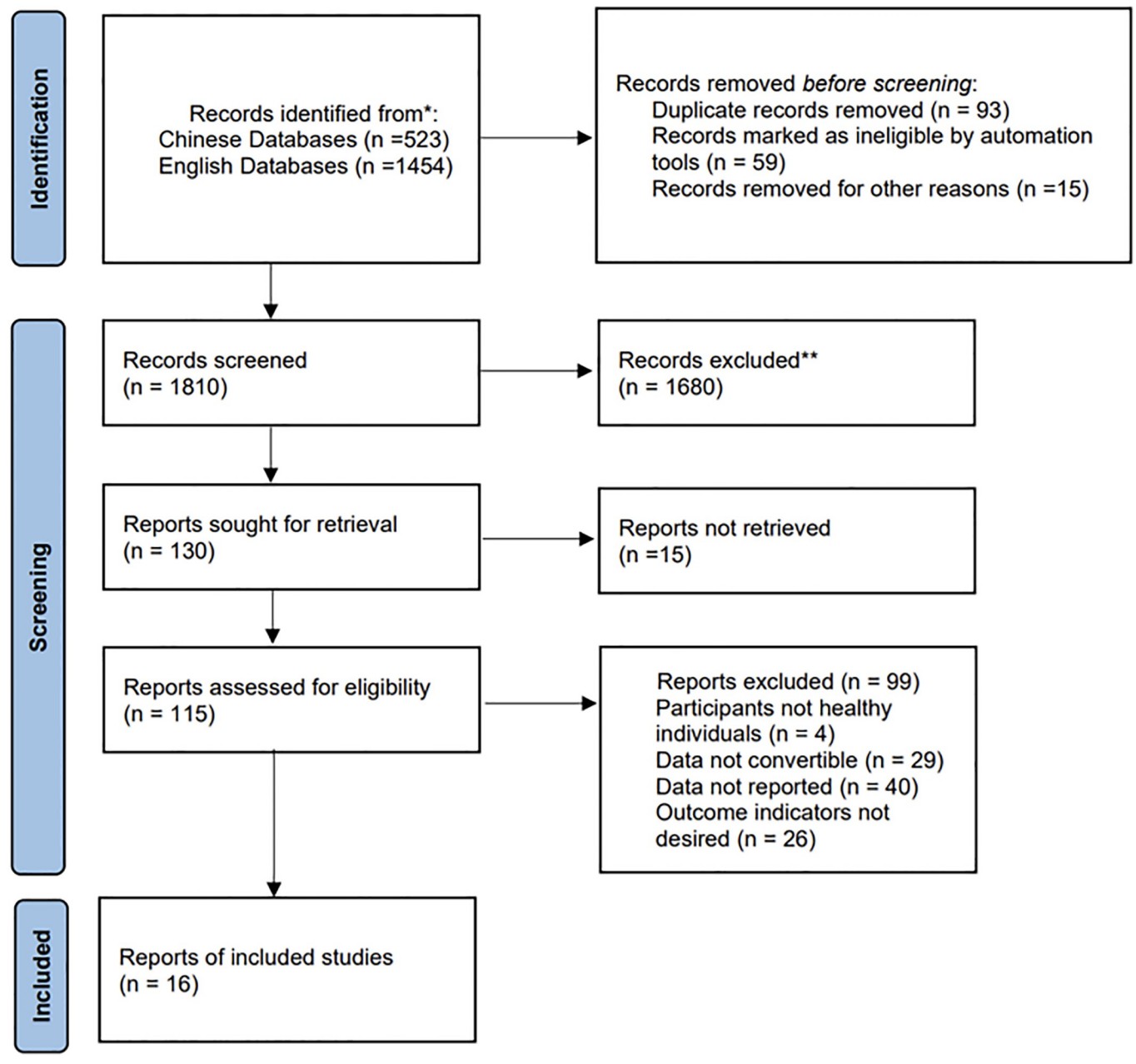

**Fig 2. PRISMA flow chart for inclusion and exclusion of studies.**

## Results

### Study characteristics

A total of 16 articles were included in this study, involving a total of 2,276 participants with ages ranging from 10 to 19 years. The intervention in the experimental group was SEM, while the control group received traditional teaching methods. The majority of studies had intervention durations of 6 weeks or 10 weeks, with sessions conducted twice a week for 55–90 minutes each (Table 1).

**Table 1. Characteristics of study participants.**

| Studies | Genders | Sample Size | | Age | | Experimental Group | | | | Control Group | Key Outcome Indicators |
|---|---|---|---|---|---|---|---|---|---|---|---|
| | | E | C | E | C | Interventions | Frequency | Duration | | Interventions | |
| Burgueo 2018 [18] | Male/ Female | 22 | 22 | 16.32 ±0.57 | 16.32 ±0.57 | SEM+ Basketball | 2/week | 55min | 6 weeks | TTM+ Basketball | ①②③ |
| Burgueo 2017 [19] | Male/ Female | 22 | 22 | 16.32 ±0.57 | 16.32 ±0.57 | SEM+ Basketball | 2/week | 55min | 6 weeks | TTM+ Basketball | ④ |
| Cuevas 2016 [20] | Male/ Female | 43 | 43 | 15.65 ±0.78 | 15.65 ±0.78 | SEM+ Volleyball | 2/week | 55min | 10 weeks | TTM+ Volleyball | ①②③④⑦ |
| Cuevas 2015 [10] | Male/ Female | 43 | 43 | 15.65 ±0.78 | 15.65 ±0.78 | SEM+ Volleyball | 2/week | 55min | 10 weeks | TTM+ Volleyball | ①②③ |
| Hernandez 2020[21] | Male/ Female | 54 | 43 | 13.16 ±0.46 | 13.54 ±0.57 | SEM+ PE | 2/week | 50min | 6 weeks | TTM+ PE | ④ |
| Mendez 2017 [22] | Male/ Female | 24 | 70 | 11.62 ±0.79 | 11.62 ±0.79 | SEM+ football, handball, orienteering | 12 times | 60mim | / | TTM+ football、handball、orienteering | ①②③ |
| Gimenez 2013 [23] | Male/ Female | 110 | 107 | 14.2 ±1.68 | 14.2 ±1.68 | SEM+ Frisbee | 12 times | 55min | / | TTM+ Frisbee | ①②③ |
| Sun 2016[11] | Male/ Female | 30 | 30 | 16 | 16 | SEM+ Basketball | 1/week | 60min | 10 weeks | TTM+ Basketball | ①②③ |
| Viciana 2020 [24] | Male/ Female | 67 | 42 | 14–15 | 14–15 | SEM+ Volleyball | 12 times | / | / | TTM+ Volleyball | ④⑦ |
| Wallhead 2014 [25] | Male/ Female | 261 | 277 | 14.75 ±0.48 | 14.75 ±0.48 | SEM+ PE | 2-3/week | 90min | / | TTM+ PE | ⑥ |
| Wallhead 2004 [26] | Male | 25 | 26 | 14.3 ±0.48 | 14.3 ±0.48 | SEM+ PE | 8 times | 60min | / | TTM+ PE | ②⑥ |
| Franco 2021 [27] | Male/ Female | 25 | 25 | 14.61 ±0.5 | 14.61 ±0.5 | SEM+ Basketball | 2/week | 50min | 4 weeks | TTM+ Basketball | ①②③ |
| Manso 2020 [28] | Male/ Female | 46 | 64 | 10.7 ±0.6 | 10.7 ±0.6 | SEM+Goubaksport | 3/week | 45min | / | TTM+Goubaksport | ①②③ |
| Choi 2020[29] | Male/ Female | 188 | 184 | 18.53 ±0.93 | 18.57 ±1.04 | SEM+ PE | 1/week | 90min | 10 weeks | TTM+ PE | ④ |
| Xu 2022 (M) [30] | Male | 104 | 103 | 18.11 ±0.76 | 18.42 ±1.09 | SEM+ Volleyball | 2/week | 90min | 17 weeks | TTM+ Volleyball | ⑤ |
| Xu 2022 (F) [30] | Female | 104 | 103 | 18.09 ±0.70 | 18.26 ±0.73 | SEM+ Volleyball | 2/week | 90min | 17 weeks | TTM+ Volleyball | ⑤ |
| Shen 2018[31] | Male | 54 | 57 | 13.8 ±0.4 | 13.7 ±0.5 | SEM+ Basketball | 2/week | 40min | 18 weeks | TTM+ Basketball | ⑤ |

Note 1:M: Man, F:Female, E:experimental group, C:control group, SEM:Sport Education Model, TTM:Traditional Teaching Model, ①: autonomy, ②: competence, ③: relatedness, ④: intrinsic motivation, ⑤: interest, ⑥: enjoyment, ⑦: satisfaction.

## Risk of bias in the included articles

In this study, the PEDro scale was used to assess the risk of bias in the included literature (Table 2). The average score of the included literature was 5.6, indicating a moderate quality. Among them, 9 articles scored 5, 4 articles scored 6, and 3 articles scored 7.

## Meta-analysis results

**Basic psychological needs.** This study included a total of 9 articles comprising 37 studies to evaluate the impact of SEM on the basic psychological needs of adolescent students. As shown in Fig 3, the meta-analysis results indicated that, overall, SEM has a positive effect on the basic psychological needs of adolescent students (MD = 0.36, 95% CI [0.22, 0.50]). The

**Table 2. Evaluation form for literature quality.**

|  | 1 | 2 | 3 | 4 | 5 | 6 | 7 | 8 | 9 | 10 | 11 | Score |
|---|---|---|---|---|---|---|---|---|---|---|---|---|
| Burgueo 2018 [18] | 1 | 0 | 0 | 1 | 0 | 0 | 0 | 1 | 1 | 1 | 1 | 5 |
| Burgueo 2017 [19] | 1 | 1 | 1 | 1 | 0 | 0 | 0 | 1 | 1 | 1 | 1 | 7 |
| Cuevas 2016 [20] | 1 | 0 | 1 | 0 | 0 | 0 | 0 | 1 | 1 | 1 | 1 | 5 |
| Cuevas 2015 [10] | 1 | 0 | 1 | 1 | 0 | 0 | 0 | 1 | 1 | 1 | 1 | 6 |
| Hernandez 2020 [21] | 1 | 0 | 1 | 1 | 0 | 0 | 0 | 1 | 1 | 1 | 1 | 6 |
| Mendez 2017 [22] | 1 | 0 | 0 | 1 | 0 | 0 | 0 | 1 | 1 | 1 | 1 | 5 |
| Gimenez 2013 [23] | 1 | 0 | 0 | 1 | 0 | 0 | 0 | 1 | 1 | 1 | 1 | 5 |
| Sun 2016 [11] | 1 | 0 | 0 | 1 | 0 | 0 | 0 | 1 | 1 | 1 | 1 | 5 |
| Viciana 2020 [24] | 1 | 1 | 1 | 1 | 0 | 0 | 0 | 1 | 1 | 1 | 1 | 7 |
| Wallhead 2014 [25] | 1 | 0 | 0 | 1 | 0 | 0 | 0 | 1 | 1 | 1 | 1 | 5 |
| Wallhead 2004 [26] | 1 | 0 | 1 | 1 | 0 | 0 | 0 | 1 | 1 | 1 | 1 | 6 |
| Franco 2021 [27] | 1 | 0 | 0 | 1 | 0 | 0 | 0 | 1 | 1 | 1 | 1 | 5 |
| Manso 2020 [28] | 1 | 0 | 0 | 1 | 0 | 0 | 0 | 1 | 1 | 1 | 1 | 5 |
| Choi 2020 [29] | 1 | 1 | 1 | 1 | 0 | 0 | 0 | 1 | 1 | 1 | 1 | 7 |
| Xu 2022 [30] | 1 | 0 | 0 | 1 | 0 | 0 | 0 | 1 | 1 | 1 | 1 | 5 |
| Shen 2018 [31] | 1 | 0 | 0 | 1 | 0 | 0 | 1 | 1 | 1 | 1 | 1 | 6 |

studies exhibited moderate heterogeneity ($I^2$ = 73%) and were statistically significant (P<0.01). Subgroup analysis results revealed that SEM has varying degrees of positive effects on the basic psychological needs of adolescents in three different age groups: PRE-PHV (MD = 0.39, 95% CI [0.23, 0.56], $I^2$ = 45%, P<0.01), MID-PHV (MD = 0.22, 95% CI [0.01, 0.42], $I^2$ = 82%, P<0.05), and POST-PHV (MD = 1.27, 95% CI [0.79, 1.74], $I^2$ = 0%, P<0.01).

**Intrinsic motivation.** This study included a total of 10 articles comprising 14 studies to assess the changes in intrinsic motivation among adolescent students after SEM interventions. As shown in Fig 4, the meta-analysis results indicated that, overall, SEM has a positive effect on adolescent student's intrinsic motivation (MD = 0.75, 95% CI [0.58, 0.93]). The studies exhibited moderate heterogeneity ($I^2$ = 59%) and were statistically significant (P<0.01). Under the inclusion criteria, all participants in the included literature focusing on intrinsic motivation as an outcome measure were 13 years of age or older. Since subgroup analysis was conducted only for the MID-PHV and POST-PHV subgroups, the subgroup analysis results revealed that SEM has varying degrees of positive effects on the intrinsic motivation of adolescent students in the MID-PHV subgroup (MD = 0.86, 95% CI [0.62, 1.11], $I^2$ = 68%, P<0.01) and the POST-PHV subgroup (MD = 0.56, 95% CI [0.40, 0.72], $I^2$ = 0%, P<0.01).

**Reporting bias.** The majority of effect sizes for basic psychological needs and intrinsic motivation were distributed around the central line and exhibited symmetrical distribution, indicating a low risk of study bias (Fig 5).

## Discussion

### Basic psychological needs

Overall, SEM is an effective approach to enhancing the basic psychological needs of adolescent students. Subgroup analysis results indicate that SEM has an age effect on improving the basic psychological needs of adolescents, with the highest effect observed during the POST-PHV stage (MD = 1.27, P<0.01), followed by the PRE-PHV stage (MD = 0.39, P<0.01), and a slight improvement in the MID-PHV stage (MD = 0.22, P<0.05).

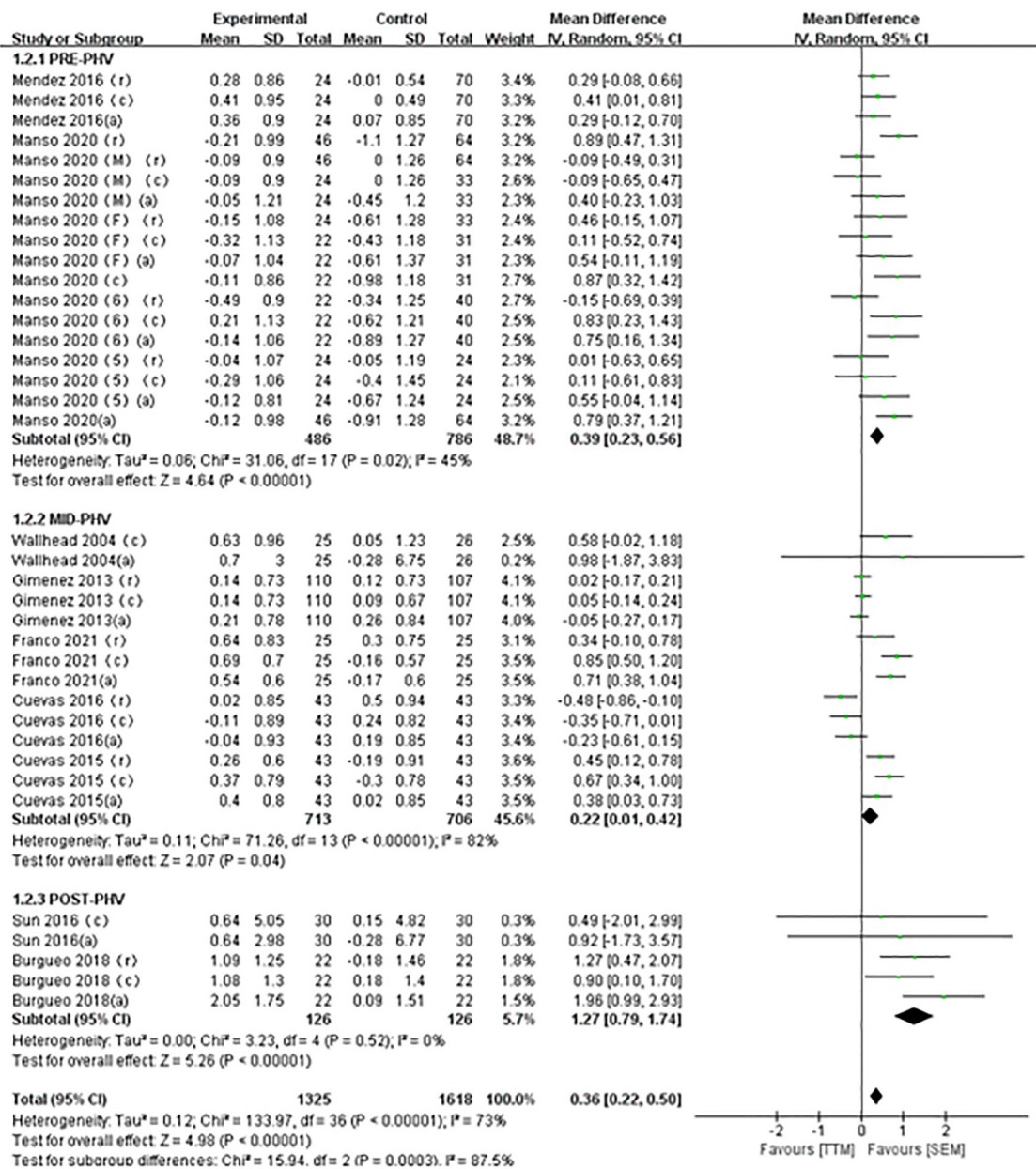

**Fig 3. Forest plot of basic psychological needs.**

The overall effect of this study is consistent with the majority of research findings on the impact of SEM on basic psychological needs [25,32]. This is because SEM creates a favorable learning environment for students, providing them with more choices and practice opportunities in physical education classes. Students have the right to establish game rules while also needing to understand and execute strategies. Additionally, during adolescence, students are increasingly influenced by their peers. By forming fixed teams that last throughout the "season," students collaborate sincerely to face various difficulties and challenges, thereby promoting relationships [33].

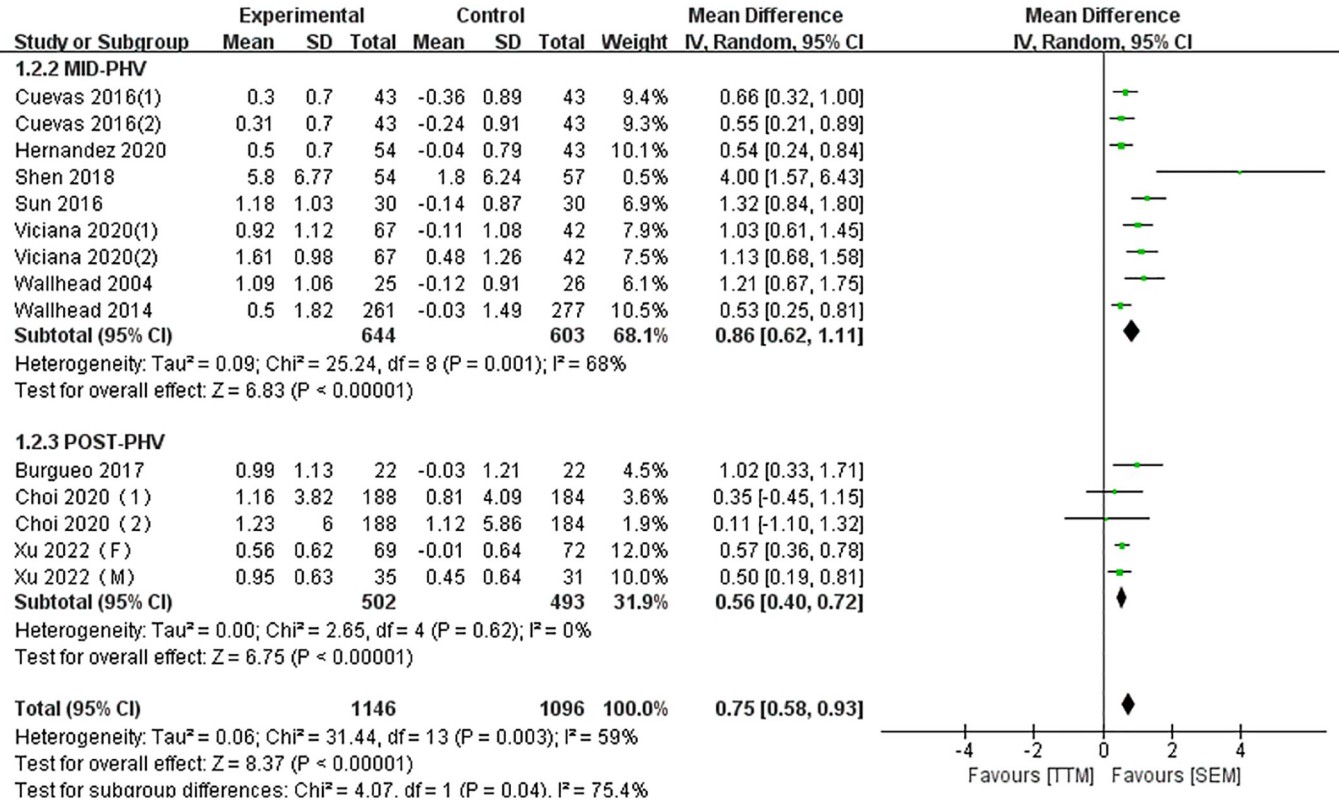

**Fig 4. Forest Plot of intrinsic motivation.**

The older the adolescent students, the higher their basic psychological needs [34], and there is an age effect in the satisfaction of basic psychological needs under SEM intervention. Firstly, the improvement of cognitive abilities is an important factor contributing to this result. With age, adolescent students have a deeper understanding of their own needs and know better how

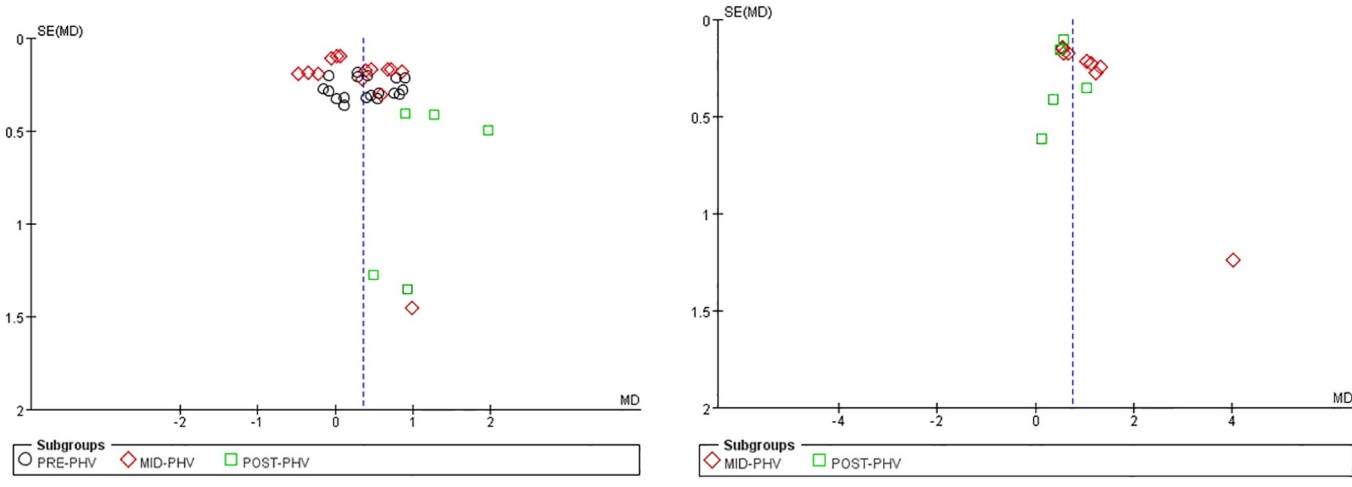

**Fig 5. Funnel plot of basic psychological needs and intrinsic motivation.**

to fulfill them. Therefore, they are more receptive to and understanding of new teaching methods, providing a foundation for the implementation of SEM. In comparison, POST-PHV students are relatively more mature and stable. Besides their desire for self-expression, they have clearer goals and pay more attention to future planning. Through SEM, adolescent students experience different roles such as coaches, referees, and record keepers, enabling them to discover their interests and potentials, which is beneficial for future career choices [28].

Secondly, the specific relationship between the developmental characteristics of adolescence and basic psychological needs provides a good entry point for SEM. During adolescence, there are changes in physical stature and physical functioning. Physiologically, adolescents experience changes in body size, significant development in muscle and skeletal systems, increased activity of sebaceous glands leading to more hair growth, and the development of the prefrontal cortex and parietal cortex in the brain, which promote learning, thinking, and decision-making abilities [35]. Psychologically, students experience enhanced self-awareness and self-esteem and become more sensitive to their bodies and appearance. These changes lead adolescents to seek autonomy and decision-making power, desire recognition in their academic and personal lives, and crave attention and support from family and friends [36]. In SEM, teachers can provide students with more autonomy and opportunities for self-expression, encouraging them to think independently. In a team-based setting, students need to contribute constructive opinions for mutual development, thus providing social and emotional support [37,38].

Lastly, the teacher's teaching style, understanding of students, and comprehension of the instructional model are important factors influencing teaching effectiveness. Teachers should have a thorough understanding of the characteristics of students at different stages of adolescence and employ different teaching methods to stimulate students' learning interests, respect their differences, and adapt to their rapidly changing learning needs [39]. The successful implementation of an instructional model relies on a solid understanding and familiarity with the model. Therefore, it is common to provide training for teachers before implementing the instructional model, and the duration of the model should be no less than 20 class hours [40], This is also a primary reason for the heterogeneity observed in the results of this study.

## Intrinsic motivation

Overall, SEM can effectively enhance the intrinsic motivation of adolescents. Subgroup analysis results indicate that SEM has an age effect on improving intrinsic motivation in adolescents, and the effectiveness of improvement decreases with increasing age (MID-PHV (MD = 0.86, P<0.01) > POST-PHV (MD = 0.56, P<0.01)).

The overall effect of this study is consistent with the majority of previous studies [41,42], Intrinsic motivation is positively associated with students' effort and interest in physical education classes. The use of Sport SEM makes physical education classes more enjoyable, and students express feelings of "happiness" and "anticipation" towards participating in physical activities. This reflects an increase in students' practical motivation towards physical education classes. Additionally, longer seasons contribute to team cohesion, and this positive learning environment can promote higher levels of motivation [43].

The level of intrinsic motivation tends to decrease with age, and therefore, the effect of SEM intervention on adolescent students' intrinsic motivation also shows a diminishing trend with increasing age. The main reasons for this phenomenon are as follows: Firstly, as adolescents grow older, their personalities and values gradually stabilize, and they become more focused on external achievements. Their response to external stimuli gradually weakens, leading to a corresponding decline in intrinsic motivation [13]. Additionally, students face increasing pressure and expectations as they age. POST-PHV students face dual pressures of academics and

career choices, which can lower their intrinsic motivation to participate in SEM. In contrast, MID-PHV students are more focused on opportunities for self-development and are more easily influenced by the external environment, making them more enthusiastic about new teaching models like SEM [44].

Furthermore, the relationship between the developmental characteristics of adolescence and intrinsic motivation can explain the age effect observed in SEM. Adolescence is a critical period for students' physiological development, and due to the reorganization of neural circuits, their cognitive and thinking abilities improve, triggering intrinsic motivation toward knowledge, creativity, and self-fulfillment. However, as they grow older, the rate of brain development slows down, and intrinsic motivation gradually weakens. This directly affects their participation in sports and the acquisition of sports skills. With reduced enthusiasm for physical education, students may struggle to master sports skills and find it challenging to fully engage in physical activities, thereby significantly diminishing the effectiveness of SEM implementation [34].

Lastly, the role of basic psychological needs in the development and maintenance of intrinsic motivation is an important factor contributing to the age effect of intrinsic motivation in adolescent students under SEM intervention. Adolescent students are more likely to develop rebellious attitudes such as truancy and disengagement from studying, and traditional school environments are unable to adequately meet their basic psychological needs. The implementation of SEM addresses this issue by providing opportunities for student expression and autonomous choice in the classroom, which can enhance students' learning interests. The display of competence is a source of student confidence, and the inability to fulfill competence needs is a significant factor in the reduction of intrinsic motivation. During adolescence, students increasingly value their social status among peers and desire to establish emotional support with others. Therefore, the age effect of social relatedness needs should not be overlooked in its impact on intrinsic motivation [18,34].

## Conclusions

The sport education model is an effective approach to enhancing the basic psychological needs and intrinsic motivation of adolescent students. However, it exhibits age effects among students at different developmental stages. Specifically, in terms of enhancing basic psychological needs, the model has the greatest impact on POST-PHV students, followed by PRE-PHV students, while the improvement effect is relatively lower for MID-PHV students. The enhancement effect on intrinsic motivation diminishes with increasing age.

## Limitations and prospects

This study has a limited number of included literature, resulting in the analysis of intrinsic motivation groups being limited to MID-PHV and POST-PHV students. The included literature mostly consists of non-randomized controlled experiments, and blinding was not implemented, which may introduce bias due to researchers' subjective intentions. Additionally, the intervention periods in the included literature ranged from 4 weeks to 18 weeks, making it difficult to determine the optimal implementation period for the sport education model. Based on the findings of this study, we plan to extend our research by recruiting adolescent students in PRE-PHV, MID-PHV, and POST-PHV for additional experimental investigations. This expanded approach will enable us to explore the impact of implementing various physical education models among different age groups, providing a more comprehensive understanding of the advantages and limitations associated with these models.

## Supporting information

**S1 Checklist. PRISMA 2020 checklist.**
(DOCX)

**S1 File. Characteristics of study participants.**
(XLSX)

**S2 File. Data extraction.**
(XLSX)

**S1 Data. Evaluation form for literature quality.**
(XLSX)

## Acknowledgments

Thank you to all the researchers who have contributed to this study.

## Author Contributions

**Conceptualization:** Duanying Li.

**Data curation:** Jing Dai, Jiayong Chen, Yezi Li.

**Formal analysis:** Jiayong Chen, Yezi Li.

**Funding acquisition:** Duanying Li.

**Investigation:** Yuhuan Chen, Min Lu.

**Methodology:** Zijing Huang, Min Lu.

**Project administration:** Yezi Li.

**Resources:** Zijing Huang, Yuhuan Chen.

**Software:** Jing Dai, Jiayong Chen, Jiancai Chen.

**Supervision:** Yuhuan Chen, Jian Sun, Duanying Li, Jiancai Chen.

**Validation:** Yuhuan Chen, Jian Sun.

**Writing – original draft:** Jing Dai.

**Writing – review & editing:** Zijing Huang, Jian Sun, Min Lu, Jiancai Chen.

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
