## [Decision Letter · Decision Letter 0]

4 Jan 2024

PONE-D-23-40957Age-effects of sport education model on basic psychological needs and intrinsic motivation of adolescent students:A meta-analysisPLOS ONE

Dear Dr. Chen,

Thank you for submitting your manuscript to PLOS ONE. After careful consideration, we feel that it has merit but does not fully meet PLOS ONE’s publication criteria as it currently stands. Therefore, we invite you to submit a revised version of the manuscript that addresses the points raised during the review process.

We look forward to receiving your revised manuscript.

Kind regards,

Henri Tilga, PhD

Academic Editor

PLOS ONE

Journal Requirements:

Additional Editor Comments:

The Reviewers have provided several useful comments to increase the quality of this manuscript. Please carefully follow all the comments made by the Reviewers and revise the manuscript accordingly.

Reviewers' comments:

Reviewer's Responses to Questions

**Comments to the Author**

1. Is the manuscript technically sound, and do the data support the conclusions?

Reviewer #1: Yes

Reviewer #2: Partly

2. Has the statistical analysis been performed appropriately and rigorously? 

Reviewer #1: Yes

Reviewer #2: Yes

3. Have the authors made all data underlying the findings in their manuscript fully available?

Reviewer #1: Yes

Reviewer #2: Yes

4. Is the manuscript presented in an intelligible fashion and written in standard English?

Reviewer #1: Yes

Reviewer #2: Yes

5. Review Comments to the Author

Reviewer #1: The study presented is interesting and reveals interest for readers and initiative on the part of the authors. It is understood that there is care in the writing and in the results presented.

the next step shoul be to perform an experimental study in these age group and variables.

Reviewer #2: The title should be written as a systematic review and meta-analysis.

What is the novelty of the work, and does the study add to our knowledge?

All abbreviations should be described for the first use.

All verbs in the method and results section should be revised. In some cases, the future form of the verb is utilized.

Keywords should be based on the MeSH.

The search date should be added in the method section of the abstract. In addition, searching the literature needs updating.

The introduction is too long.

Please add the search strategy.

What were the limitations of your work and the recommendations for future studies?

6. PLOS authors have the option to publish the peer review history of their article (what does this mean?). If published, this will include your full peer review and any attached files.

Reviewer #1: No

Reviewer #2: No

---

## [Author Response · Author response to Decision Letter 0]

9 Jan 2024

Responds to the reviewer’s comments:

Responds to the Reviewer #1: 

Reviewer's Comment 1: 

The study presented is interesting and reveals interest for readers and initiative on the part of the authors. It is understood that there is care in the writing and in the results presented.

the next step should be to perform an experimental study in these age group and variables.

Response to Comment 1：

Thank you for acknowledging our study and providing valuable suggestions for future research. We are delighted to hear that you find our research interesting and believe it has the potential to captivate readers. We also appreciate your recognition of the effort we have invested in writing and presenting our findings. This recognition holds great personal significance for me, as it strengthens my confidence in my pursuit of learning and scientific research.

We have thoroughly considered your suggestions and made revisions to the "Limitations and Prospects" section of the manuscript. We are genuinely interested in your suggestions, and going forward, we plan to create experimental conditions and design studies to further explore this direction. Our aim is to make a more comprehensive contribution to the existing knowledge in this field.

Responds to the Reviewer #2: 

Reviewer's Comment 1: 

The title should be written as a systematic review and meta-analysis.

Response to Comment 1：

We sincerely appreciate your suggestion regarding the title and value your valuable feedback on our manuscript.

After careful consideration of your feedback and internal discussions, we believe that your suggestions are absolutely correct. As such, we have updated the title of the manuscript to " Age-effects of sport education model on basic psychological needs and intrinsic motivation of adolescent students: A systematic review and meta-analysis " 

Reviewer's Comment 2: 

What is the novelty of the work, and does the study add to our knowledge?

Response to Comment 2：

Thank you for your insightful question regarding the novelty of our work and its contribution to existing knowledge. We appreciate the opportunity to address this important aspect of our study.

Basic psychological needs and intrinsic motivation are closely related to students' physical participation, self-awareness, learning interest and enthusiasm, and social adaptability in physical education. The Sport Education model (SEM), widely used in physical education, is an effective approach to enhancing students' basic psychological needs and intrinsic motivation. Currently, there is a considerable amount of research exploring the implementation effects of the SEM. However, the obtained results are inconsistent and overlook the variable of how basic psychological needs and intrinsic motivation may change with age. In order to identify the optimal implementation period of the SEM and enhance teaching effectiveness, we believe it is necessary to investigate the impact of age on its outcomes.

Therefore, we conducted a systematic review and meta-analysis of the topic, which allows us to comprehensively synthesize and analyze existing literature. By aggregating findings from multiple studies, we can provide a more robust and comprehensive assessment of the effects of the SEM on basic psychological needs and intrinsic motivation. This approach adds to our knowledge by providing a comprehensive summary of existing evidence and identifying potential gaps or areas for future research.

Reviewer's Comment 3: 

All abbreviations should be described for the first use.

Response to Comment 3：

Thank you for your valuable feedback regarding the use of abbreviations in our manuscript. We have conducted a thorough examination of the entire document and have ensured that all abbreviations are described upon their first use. We sincerely apologize for any confusion that may have arisen from our initial oversight.

We would like to express our deep appreciation for bringing this matter to our attention. Your guidance and attention to detail have been instrumental in improving the clarity and comprehensibility of our work. We recognize the importance of ensuring clarity and facilitating reader understanding, and we will make every effort to avoid similar oversights in the future.

Reviewer's Comment 4: 

All verbs in the method and results section should be revised. In some cases, the future form of the verb is utilized.

Response to Comment 4：

Thank you for your feedback. We sincerely apologize for any errors in verb tense usage in the method and results section. We have thoroughly reviewed these sections and have made the necessary revisions to rectify the issue.

We fully agree with your point that, although English is not our native language, language should not be a barrier to our research and translation. We understand the importance of using the correct verb tense and have taken steps to rectify the issue. Your guidance has been invaluable in improving the quality of our manuscript.

Reviewer's Comment 5: 

Keywords should be based on the MeSH.

Response to Comment 5：

Thank you for your valuable feedback regarding the keywords used in our manuscript. In line with your suggestion, we conducted a search in MeSH using the terms "sport education model, adolescent students, basic psychological needs, intrinsic motivation." However, the search yielded no results with the message "No items found.".

In light of this limitation, we have chosen alternative keywords that best describe the key concepts and topics covered in our study. These keywords were selected based on our understanding of the field and relevant literature.

Reviewer's Comment 6: 

The search date should be added in the method section of the abstract. In addition, searching the literature needs updating.

Response to Comment 6：

Thank you for your valuable guidance. We sincerely appreciate your suggestion to include the search date in the method section of the abstract. We believe your suggestion is absolutely correct. We have now added the search date to the methods section of the abstract. 

Furthermore, we appreciate your comment on updating the literature search. We understand the importance of staying informed about the latest research. Therefore, on January 7, 2024, we conducted a new search on four databases, namely PubMed, Web of Science, Scopus, and China National Knowledge Infrastructure (CNKI). This was done to ensure that our findings were based on the most up-to-date information available. We have also updated the flowchart of the literature screening process. However, after conducting the literature screening, we did not identify any studies that met our inclusion criteria. The updated literature screening flowchart will be provided as an attachment.

Reviewer's Comment 7: 

The introduction is too long.

Response to Comment 7：

Thank you for your feedback regarding the length of the introduction in our manuscript. We appreciate your input and understand your concern.

After carefully considering your comment, we have revised the introduction to make it more concise and focused. We have condensed the background information while still providing the necessary context for our study. By doing so, we aim to improve the overall flow and readability of the manuscript.

Reviewer's Comment 8: 

Please add the search strategy.

Response to Comment 8：

Thank you for your valuable suggestion. We greatly appreciate your feedback. In response to your comment, we have incorporated a visual representation of the literature search strategy in the form of an attached image. The image attached provides a clear overview of the search terms used in the PubMed database. It serves as an illustrative example, summarizing the keywords utilized in the search strategy.

We believe that the inclusion of the search strategy will enhance the transparency and reproducibility of our research. 

Reviewer's Comment 9: 

What were the limitations of your work and the recommendations for future studies?

Response to Comment 9：

Thank you for your insightful question. We are honored to have the opportunity to discuss the limitations of our manuscript and provide recommendations for future research.

We have provided a description of the limitations and future prospects in the " Limitations and Prospects " section of our manuscript. Specifically, the limitations of our study are as follows: This study has a limited number of included literature, resulting in the analysis of intrinsic motivation groups being limited to pre-peak height velocity(MID-PHV) and post-peak height velocity(POST-PHV) students. The included literature mostly consists of non-randomized controlled experiments, and blinding was not implemented, which may introduce bias due to researchers' subjective intentions. Additionally, the intervention periods in the included literature ranged from 4 weeks to 18 weeks, making it difficult to determine the optimal implementation period for the sport education model. Based on the findings of this study, we suggest to extend our research by recruiting adolescent students in PRE-PHV, mid-peak height velocity(MID-PHV), and POST-PHV for additional experimental investigations. This expanded approach will enable us to explore the impact of implementing various physical education models among different age groups, providing a more comprehensive understanding of the advantages and limitations associated with these models.

---

## [Decision Letter · Decision Letter 1]

15 Jan 2024

Age-effects of sport education model on basic psychological needs and intrinsic motivation of adolescent students: A systematic review and meta-analysis

PONE-D-23-40957R1

Dear Dr. Chen,

We’re pleased to inform you that your manuscript has been judged scientifically suitable for publication and will be formally accepted for publication once it meets all outstanding technical requirements.

Kind regards,

Henri Tilga, PhD

Academic Editor

PLOS ONE

Additional Editor Comments (optional):

Reviewers' comments:

Reviewer's Responses to Questions

**Comments to the Author**

1. If the authors have adequately addressed your comments raised in a previous round of review and you feel that this manuscript is now acceptable for publication, you may indicate that here to bypass the “Comments to the Author” section, enter your conflict of interest statement in the “Confidential to Editor” section, and submit your "Accept" recommendation.

Reviewer #1: All comments have been addressed

Reviewer #2: All comments have been addressed

2. Is the manuscript technically sound, and do the data support the conclusions?

Reviewer #1: Yes

Reviewer #2: Partly

3. Has the statistical analysis been performed appropriately and rigorously? 

Reviewer #1: Yes

Reviewer #2: Yes

4. Have the authors made all data underlying the findings in their manuscript fully available?

Reviewer #1: Yes

Reviewer #2: Yes

5. Is the manuscript presented in an intelligible fashion and written in standard English?

Reviewer #1: Yes

Reviewer #2: Yes

6. Review Comments to the Author

Reviewer #1: the authors have adressed all suggestions made in the previous review, so its my oppinion that is acceptable for publication

Reviewer #2: Thank you for the correction of authors regarding my comments. All of my comments have been addressed satisfactorily.

7. PLOS authors have the option to publish the peer review history of their article (what does this mean?). If published, this will include your full peer review and any attached files.

Reviewer #1: No

Reviewer #2: No
